# Dietary Patterns and New-Onset Type 2 Diabetes Mellitus in Evacuees after the Great East Japan Earthquake: A 7-Year Longitudinal Analysis in the Fukushima Health Management Survey

**DOI:** 10.3390/nu14224872

**Published:** 2022-11-17

**Authors:** Enbo Ma, Tetsuya Ohira, Hiroyuki Hirai, Kanako Okazaki, Masanori Nagao, Fumikazu Hayashi, Hironori Nakano, Yuriko Suzuki, Akira Sakai, Atsushi Takahashi, Junichiro J. Kazama, Hirooki Yabe, Masaharu Maeda, Seiji Yasumura, Hitoshi Ohto, Kenji Kamiya, Michio Shimabukuro

**Affiliations:** 1Health Promotion Center, Fukushima Medical University, Fukushima 960-1295, Japan; 2Department of Epidemiology, Fukushima Medical University School of Medicine, Fukushima 960-1295, Japan; 3Radiation Medical Science Center for Fukushima Health Management Survey, Fukushima Medical University, Fukushima 960-1295, Japan; 4Department of Diabetes, Endocrinology, and Metabolism, Fukushima Medical University School of Medicine, Fukushima 960-1295, Japan; 5Department of Physical Therapy, Fukushima Medical University School of Medical Sciences, Fukushima 960-8516, Japan; 6Mental Health, National Center of Neurology and Psychiatry, Tokyo 187-8553, Japan; 7Department of Radiation Life Sciences, Fukushima Medical University School of Medicine, Fukushima 960-1295, Japan; 8Department of Gastroenterology, Fukushima Medical University School of Medicine, Fukushima 960-1295, Japan; 9Department of Nephrology and Hypertension, Fukushima Medical University School of Medicine, Fukushima 960-1295, Japan; 10Department of Neuropsychiatry, Fukushima Medical University School of Medicine, Fukushima 960-1295, Japan; 11Department of Disaster Psychiatry, Fukushima Medical University School of Medicine, Fukushima 960-1295, Japan; 12Department of Public Health, Fukushima Medical University School of Medicine, Fukushima 960-1295, Japan; 13Research Institute for Radiation Biology and Medicine, Hiroshima, University, Hiroshima 734-8553, Japan

**Keywords:** dietary pattern, diabetes mellitus

## Abstract

**Background**: Dietary patterns may be linked to the incidence of type 2 diabetes mellitus (T2DM) after disasters. We investigated the association between dietary patterns and new-onset T2DM in evacuees of the Great East Japan Earthquake and the Fukushima Daiichi Nuclear Power Plant (FDNPP) accident. **Methods**: Among the 22,740 non-diabetic participants aged 20–89 years who completed the dietary assessment in the Fukushima Health Management Survey between July 2011 and November 2012, the incidence of T2DM was evaluated until 2018. Principal component analysis with varimax rotation was applied to derive dietary patterns based on a validated, short-form food frequency questionnaire. The identified dietary patterns were categorized as typical Japanese, juice, and meat. **Results:** The cumulative incidence of T2DM was 18.0 and 9.8 per 1000 person-years in men and women, respectively, during the follow-up period. The multiple-adjusted hazard ratio (95% confidence interval) of the highest vs. lowest quartile of the typical Japanese pattern scores for T2DM was 0.80 (0.68, 0.94; *P* for trend = 0.015) in total, 0.85 (0.68, 1.06; *P* for trend = 0.181) in men, and 0.76 (0.60, 0.95; *P* for trend = 0.04) in women. **Conclusions:** A typical Japanese dietary pattern may be associated with a reduced new-onset T2DM risk in evacuees, especially women, after the Great East Japan Earthquake and the FDNPP accident.

## 1. Introduction

There is evidence of associations between dietary behaviors or diet quality indices, food groups, single foods and beverages, specific macronutrients and micronutrients, and the incidence of type 2 diabetes mellitus (T2DM) [1,2,3]. Dietary patterns are one such factor considered to play a role in the onset or prevention of T2DM [1,2,3,4,5]. The Mediterranean diet, Dietary Approaches to Stop Hypertension (DASH) diet, and diets with high Alternate Healthy Eating Index (AHEI) scores have the potential to prevent T2DM [1,2,3,4,5]. The “healthy/prudent” and “unhealthy/Western” patterns are also regarded as being associated with a reduced and increased risk of developing T2DM, respectively. The Japanese diet differs from diets in Western countries [6]. Therefore, it is necessary to identify appropriate dietary patterns for Japanese individuals to prevent the onset of T2DM. Previous reports have indicated that the frequency of white rice intake is associated with an increased risk of T2DM, while a high-dairy, high-fruit-and-vegetable, and low-alcohol pattern [7] and a vegetable pattern [8] have been associated with a decreased risk.

After the Great East Japan Earthquake on 11 March 2011, the Fukushima Health Management Survey (FHMS) was launched to monitor residents’ health statuses in the radiation-exposed areas of the Fukushima Daiichi Nuclear Power Plant (FDNPP) accident [9]. Studies via the FHMS have reported increased cardiovascular risks such as overweightness/obesity, hypertension, and dyslipidemia in residents after the disaster [10,11,12]. We also found that the incidence rate of diabetes in FHMS participants was significantly higher than in the general Japanese population [13]. A meta-analysis reported that new-onset T2DM increases among disaster evacuees [14]. Dietary factors, including dietary patterns, are linked to the onset of T2DM [2,4,5]. However, the association between dietary factors and new-onset T2DM among disaster evacuees remains unclear. We previously identified three dietary patterns associated with cardiometabolic risk factors in evacuees of the Great East Japan Earthquake [15]. However, the longitudinal effects of dietary patterns on new-onset T2DM have not been elucidated in this population.

This study examined the association between dietary patterns and T2DM incidence in evacuees of the Great East Japan Earthquake and the FDNPP accident.

## 2. Materials and Method

### 2.1. Study Design and Study Participants

This study was part of FHMS that targeted evacuees aged 20–90 years at the time of the earthquake and FDNPP accident and was officially registered as belonging to 13 administrative districts (villages, towns, and cities) [9]. Administrative districts included evacuation and non-evacuation zones. The FHMS includes four detailed annual surveys: thyroid ultrasound examination, comprehensive health check, mental health, lifestyle survey, and pregnancy and birth survey [9]. Among the participants who received the mental health and lifestyle survey (*n* = 70,103) and those who underwent the comprehensive health check (*n* = 51,503) at baseline between July 2011 and March 2012, we selected 36,168 participants (men 14,925, women 21,243) who underwent the two surveys (Figure 1). The ethics review committee of Fukushima Medical University approved the study protocol (#29064), and all participants provided written informed consent.

### 2.2. Dietary Intake Assessment

We applied a short-form and self-administrated food frequency questionnaire (FFQ) with 19 food items to determine the participants’ food intake six months preceding the survey date, as described previously [16]. The FFQ is a validated and modified questionnaire from the Hiroshima and Nagasaki Life Span Study [17]. In a validation study of the original FFQ, food intake measured by the FFQ was moderately correlated with those measured by the 24 h recall records [17]. In the FFQ, the 19 food items included non-juice fruits, non-juice vegetables (red and orange vegetables, green vegetables, and light-colored vegetables), fruit juice, vegetable juice, meat (chicken, beef/pork, and ham/sausage), soybean products (fermented soybean, soy milk, miso soup, tofu, and boiled beans), fish (raw and cooked), dairy (milk, yogurt, and Lactobacillus drink), rice, and bread [16]. The participants replied with how often they consumed individual food items based on six response choices for frequency: none, <1 time/wk., 1–2 times/wk., 3–4 times/wk., 5–6 times/wk. or daily [16].

### 2.3. Diabetes- and Disaster-Related Variables

General participant characteristics and diabetes- or disaster-related variables were assessed using self-reporting questionnaires. Smoking status was classified into three categories: never, former, and current. Drinking status was classified as never, occasionally, or regularly. Physical activity was classified into four categories: almost every day, 2–4 times/week, once/week, and almost never.

Participants were grouped into “evacuee” or “non-evacuee.” Evacuees were defined as those from the evacuation zone or those from the non-evacuation zone who experienced living arrangements in evacuation shelters and temporary houses.

Laboratory data obtained from the participants included measurements of high-density lipoprotein cholesterol (HDL-C), low-density lipoprotein cholesterol (LDL-C), triglycerides (TG), fasting plasma glucose (FPG), and hemoglobin A1c (HbA1c). T2DM was defined as FPG level ≥ 126 mg/dL, HbA1c level ≥ 6.5%, or self-reported use of antihyperglycemic agents. Hypertension was defined as a systolic blood pressure > 140 mmHg, diastolic blood pressure > 90 mmHg, or self-reported use of antihypertensive agents. Dyslipidemia was defined as an LDL-C level ≥ 140 mg/dL, triglyceride level ≥ 150 mg/dL, HDL-C level < 40 mg/dL, or use of lipid-lowering agents. Height (in stocking feet) and weight (wearing light clothing) were measured for each participant, and body mass index (BMI) was calculated as weight (kg) divided by the square of the height (m^2^) and grouped as <23.0, ≥23.0 to <25.0, and ≥25.0 kg/m^2^ (overweight).

The Japanese version [18] of the Kessler 6 scale (K6) [19] was used to measure non-specific mental health distress. Participants were asked if they had experienced any of 6 symptoms during the preceding 30 days: ‘feeling so sad that nothing could cheer you up,’ ‘feeling nervous’, ‘feeling hopeless’, ‘feeling restless or fidgety’, ‘feeling everything was an effort’, and ‘feeling worthless’. Probable depression was defined as a K6 score ≥ 13 out of a total of 24 points, as described previously [20].

### 2.4. Statistical Analysis

We excluded participants with more than three missing FFQ answers or those with a history of diabetes mellitus. Finally, we included 22,740 individuals who had undergone both the mental health and lifestyle surveys, the comprehensive health check in FY2011, and at least one health checkup in the follow-up years (Figure 1).

The 4.1% of the participants who did not answer more than three dietary questions on the FFQ were excluded from the analysis. For the remaining participants (95.9%), we replaced the missing food item values with the sex-specific median value of the respective food item frequency [16]. For each food item frequency, we used the daily midpoint for the frequency category; for example, we assessed ‘3–4 times/week’ as 0.5 times/day [16].

We applied a principal component analysis to derive sex-specific dietary patterns from the food items. Varimax rotation was performed to obtain a simple structure with greater interpretability. We selected factor numbers mainly based on eigenvalues >1.5, scree plots, and factor interpretability of food items, with absolute factor loadings ≥0.3 to account for each component [21]. A three-factor solution described the most distinctive dietary patterns in the study population. We labeled dietary patterns as “typical Japanese”, “juice”, and “meat” based on food items with high factor loadings on each pattern (Appendix A). The eigenvalues of the typical Japanese, juice, and meat patterns were 4.15, 1.76, and 1.64, respectively, in men, and 4.01, 1.71, and 1.60, respectively, in women. The cumulative explained variance was 39.8% in men and 38.6% in women. Moreover, the Cronbach’s α coefficient for each dietary pattern with standardized variables indicated higher internal reliability of these measures: 0.797 for typical Japanese, 0.806 for juice, and 0.813 for meat patterns in men; and 0.783 for typical Japanese, 0.799 for juice, and 0.804 for meat patterns in women. We assigned each participant a pattern-specific score, which we calculated as the sum of the products of factor-loading coefficients and standardized food intake. Factor scores reflect how closely a participant’s diet resembles the identified pattern, that is, the higher the scores, the closer the resemblance of diets [22]. We categorized the dietary pattern scores into quartiles (Q1–Q4) for further analysis. In each pattern, participants in the higher quartile group consumed representative food items more frequently than those in the lower quartile group(s).

For baseline FY2011 characteristics across groups, we used the χ^2^ test for categorical variables and the analysis of variance for continuous variables. Univariate survival analysis for new-onset diabetes mellitus was performed using the Kaplan–Meier method in typical Japanese, juice, and meat dietary patterns among Q1–Q4. We also used the Cox regression model to estimate the associations between dietary patterns and the incidence of new-onset T2DM in the follow-up years—2012–2018. Hazard ratios (HR) and 95% confidence intervals (95% CI) were estimated using the first quartile of pattern scores as the reference. Multivariate regressions were conducted with adjustments for variables according to previous publications on FHMS [11,15,16]. In Model 1, the adjustment included age (continuous) and sex; in Model 2, Model 1 plus BMI (<23, ≥23–<25, ≥25 kg/m^2^); and in Model 3, Model 2 plus education, smoking (never, former, or current), current drinking (no, yes), education level (<, ≥vocational university), physical activity (<two, ≥two times/week), probable depression (K6 < 13, ≥13), change of residence (no, yes), hypertension (no, yes), and dyslipidemia (no, yes) by category. Trends were tested using the median pattern scores in quartile categories as continuous variables.

To minimize confounding effects of <40 years of onset of T2DM, which could reflect a genetic background of cardiovascular diseases (CVD) and cancer, which often induce the onset of T2DM in the elderly, we performed sensitivity analyses for participants aged 20–89 years (*n* = 19,811) or 40–74 years (*n* = 13,632) without a history of CVD or cancer in FY2011. Repeated procedures were applied to obtain dietary pattern scores and examine their associations with T2DM incidence risk. Data were analyzed using SAS statistical software (version 9.4; SAS Institute, Cary, NC, USA). *p*-values were reported as two-sided, and *p* < 0.05 was considered statistically significant.

## 3. Results

Table 1 shows the characteristics of the study participants in FY2011. The mean age of the participants was 55.9 (SD 15.7) years. Men were older and had a higher frequency of current smokers, current alcohol drinking, and regular physical activity than women, but a lower frequency of probable depression (K6 ≥ 13). The rates of cardiometabolic risks, such as obesity, hypertension, hyperglycemia, and dyslipidemia, were higher in men than in women, except for LDL-C level. No significant differences in dietary pattern scores were observed between the men and women.

The cumulative incidence of T2DM was 18.0/1000 (731 out of 40,688 person-years) in men and 9.8/1000 (717 out of 73,082 person-years) in women during the follow-up period between 2011 and 2018 (Table 2). The Kaplan–Meier curves were shown in all, men, and women for typical Japanese, juice, and meat dietary patterns (Figure 2). The incidence of new-onset T2DM was inversely associated with typical Japanese dietary pattern scores in both men and women, with age-sex (Model 1) and plus BMI (Model 2) adjustment. When further adjusted for smoking, drinking, education level, physical activity, K6 (<13, ≥13), change of residence, overweight, and dyslipidemia (Model 3), the association was significant in women but not in men (Table 3). No significant associations were observed between juice and meat pattern scores in either men or women.

In the sensitivity analysis for participants aged 20–89 years without diabetes mellitus, CVD, and cancer in FY2011, the HRs in the highest vs. lowest quartile of typical Japanese pattern scores were significant in Model 3 in total and women but not in men (Appendix A). We also conducted a sensitivity analysis of participants aged 40–74 years, accounting for 68.8% of the total. When limited to participants aged 40–74 and without diabetes mellitus, CVD, and cancer in FY2011, similar significant HRs of the typical Japanese dietary pattern were observed in the highest quartile in total and women but not in men; the *P* for the trend in Model 3 was attenuated not to be significant in women (Appendix A). Similar to the main results, we observed significant associations between the juice or meat consumption pattern and T2DM risk in the sensitivity analysis.

## 4. Discussion

We evaluated the association between dietary patterns and new-onset T2DM in evacuees of the Great East Japan Earthquake and obtained two major findings. First, we identified three dietary patterns associated with new-onset T2DM. The typical Japanese, juice, and meat patterns were similar to those associated with cardiometabolic risks, such as being overweight, dyslipidemia, and hypertension [15]. Second, we found that the typical Japanese pattern was protective against new-onset T2DM in both men and women after adjusting for confounding factors (2). The highest vs. the lowest quintile of typical Japanese pattern scores had a 22% decrease in men and a 30% decrease in women with new-onset T2DM during the years of follow-ups (Model 2).

Based on 24 h dietary records from the Japan National Health Nutrition Survey (NHNS), three main dietary patterns were identified to stratify cardiovascular risk [8,23,24]. The three patterns included (1) vegetable/traditional Japanese/plant food and fish; (2) high-bread and low-rice/bread and dairy/Westernized; and (3) high-meat and low-fish/meat and fat/animal food and oil [8,23,24]. The patterns identified in our study were similar to those of the NHNS. The typical Japanese diet pattern in our study was similar to the traditional Japanese diet, and the juice and meat patterns were closer to the Western-style diet in the NHNS [6,25]. The NHNS reported that the ‘plant food and fish’ pattern score declined while the ‘bread and dairy’ and ‘animal food and oil’ pattern scores increased between 2003 and 2015. This finding suggests a transition from the traditional Japanese pattern to a Westernized pattern during that period [24].

It should be noted that a simplified name for a dietary pattern does not always accurately capture the full range of input variables that significantly contribute to the variance [26]. The identified dietary patterns might differ for FFQs with different food items or surveys in different populations. Nevertheless, although our study FFQ was a short form, the similar coverage of the main food groups could clarify the stability of dietary patterns in this study population.

Similar to previous studies [7,27,28], the NHNS 2012 observed that the vegetable dietary pattern was associated with a decreased prevalence of high HbA1c levels [8]. Meanwhile, we found an inverse association between typical Japanese dietary pattern scores and T2DM and HbA1C in total and women (data not shown). The incidence of T2DM could be lower in participants with a high intake of plant-based foods due to a reduced risk of obesity [29]. In contrast, saturated fats in animal foods are harmful to pancreatic β-cells and insulin-sensitizing tissues such as skeletal muscles and the liver via so-called lipotoxicity [30,31]. Plant-based foods may benefit the control of insulin resistance by reducing adiposity [32]. A recent systematic review indicated that a high intake of fruits or “vegetables and fruits,” but not vegetables alone, is associated with a reduction in T2DM onset [33]. A Korean study reported that a diet high in vegetables, mushrooms, seaweeds, fruits, and soy products and low in fatty fish and high-fat meat might protect against T2DM development [34]. A Chinese population showed a decrease in the risk of impaired blood glucose control with a similar food pattern [35]. Soybean foods are rich in folate, which decreases homocysteine and the risk of T2DM [36]; the Japanese diet, like the Mediterranean diet, is associated with the intake of antioxidant vitamins, minerals, and dietary fiber and ω-3 fatty acids [29]. On the other hand, low vegetable intake was associated with skipping meals, alcohol intake, and smoking in an NHNS 2003 report [37].

We found that the typical Japanese pattern was protective for new-onset T2DM in both men (HR 0.78, 95% CI 0.63–0.97) and women (HR 0.70, 0.56–0.88) after adjusting for age and BMI (Model 2). However, after adjusting for smoking, current drinking, education level, physical activity, probable depression, change of residence, hypertension, and dyslipidemia, the typical Japanese pattern was protective in women but not men. The current study could not explain the reason for these sex differences. A higher vegetable diet intake in women than in men and other lifestyle risks, that is, a healthier lifestyle in women than in men, might contribute to the sex difference in the associations between dietary patterns and T2DM onset [38,39,40]. Future studies are required to elucidate the mechanisms underlying sex differences in the associations between dietary patterns and new-onset T2DM in the current population.

A Japanese cohort study observed that women with daily consumption of soft drinks vs. non-consumers had 1.79 and 2.10 times higher T2DM onset at five and ten years of follow-up, respectively, but intake of 100% fruit juice and vegetable juice was not associated with T2DM onset [41]. Milk and dairy products have significant anti-inflammatory effects in healthy and metabolically abnormal subjects [42]. Yogurt intake is associated with a reduced risk of T2DM and metabolic syndrome in the general population [43,44]. The Health Professionals Follow-Up Study reported that red meat consumption was associated with an increased risk of T2DM [45]. However, we did not observe significant associations with T2DM risk with respect to juice or meat consumption patterns in this study. Nevertheless, the associations and/or consumption recommendations of diets, particularly for juice patterns (e.g., sugar-based beverages) or meat patterns (e.g., processed meat), were inconsistent in Asian populations, and further relevant studies are necessary. In addition, due to the Great East Japan Earthquake of March 2011, residents along the radiation-disclosed areas were evacuated. Therefore, changes in dietary patterns may have impacted their health and were associated with changes in lifestyle [11]. The perceptions of a healthy food environment, reflecting an individual’s intentions and motivations in lowering T2DM onset and prevalence, need to be enhanced [46].

The strengths of our study were that it included a large sample with a long follow-up period for the accumulated incidence risk of T2DM and that the associations were measured through an identified cohort. Second, the association between dietary patterns and T2DM onset might be modified by BMI [47,48]. In the multivariate regression for BMI and other factor adjustments, significant associations indicated that the typical Japanese dietary pattern score was negatively correlated with T2DM onset independently. Third, the sensitivity analysis was limited to study participants without CVD or cancer or to participants aged 40–74 without CVD or cancer in FY2011. The analysis showed consistent inverse associations between the typical Japanese pattern and T2DM onset.

This study has a few limitations. First, the FHMS included mostly evacuees due to the disaster in March 2011, and response rates remained less than 27%; thus, the results might not be generalizable to the whole prefecture or country’s population. The characteristics of the population were slightly different from those reported in the national nutritional survey. For example, in participants aged more than 20 years in the NHNS 2011, the proportion of overweight participants was 25.5%, and it was 29.8% in the FHMS; the SBP/DBP was 135.0/81.2 mmHg in men and 128.0/76.6 mmHg in women in the NHNS, while it was 130.9/79.8 mmHg in men and 124.7/74.8 mmHg in women in the FHMS; the LDL-C was 118.3 mg/dL in men and 121.5 mg/dL in women, while it was 122.8 mg/dL in men and 125.1 mg/dL in women in the FHMS. The incidence of new-onset T2DM was 9.6 per 1000 person-years (95%CI 8.3–11.1) in Japanese-pooled studies [49], while it was 19.6 per 1000 person-years [13] in this study, suggesting that the incidence was significantly higher in this cohort of participants. Second, as the FFQ was a semi-questionnaire adapted from a previous study without information on the portion sizes of food intake [17], we could not compute the food nor nutrient amounts and energy for adjustment, which may help elucidate the underlying mechanisms of T2DM [50]. In addition, the 19 food group items may not have certain correlations between specific foods [51]. It may be difficult to classify broader types of food intake when deriving dietary patterns with minimal loss of original information. For example, we did not have information on soft drinks. We also did not observe significant associations between meat patterns and T2DM onset through simple food groups, in contrast to the observations of other studies [4,52]. Third, we only computed the baseline dietary pattern scores from the FFQ surveys. Thus, we could not clarify whether participant residence, employment, or dietary habits affected the associations measured over the years. In a previous study, we reported the same three dietary patterns for three waves of food frequency surveys between 2011 and 2013 [15]. In this study, based on the dietary patterns identified by the FFQ in the 2011 survey, we calculated the standardized scores for participants who also attended the 2012 and 2013 surveys. We added the reclassifications of participants’ dietary pattern scores in the regression models for adjustment, and we observed the same inverse associations between the typical Japanese dietary patterns and T2DM risk (Appendix A). Finally, residual confounding is possible, as in any observational study. The pathophysiology of T2DM onset differs between obese and non-obese (lean) individuals [31]. Recently, the clustering-based classification of T2DM has attracted the attention of researchers in clinical diabetology [31,53]. Based on the classification, T2DM can be categorized into cluster 2 (severe insulin-deficient diabetes [SIDD]), cluster 3 (severe insulin-resistant diabetes [SIRD]), cluster 4 (mild obesity-related diabetes [MOD]), and cluster 5 (mild age-related diabetes [MARD]) [31,53]. It can be useful to assume the mechanisms in future studies on how typical Japanese dietary patterns could prevent the onset of T2DM in individuals with obesity or lean phenotypes and those with diabetes clustering (insulin resistance or β-cell dysfunction).

In conclusion, a typical Japanese dietary pattern was associated with a reduced risk of T2DM in evacuees of the Great East Japan Earthquake, and the benefit of a typical Japanese diet pattern was observed preferentially in women. Further studies are needed to elucidate the mechanisms underlying the benefits of the typical Japanese diet by incorporating detailed information on sex disparities.

## Figures and Tables

**Figure 1 nutrients-14-04872-f001:**
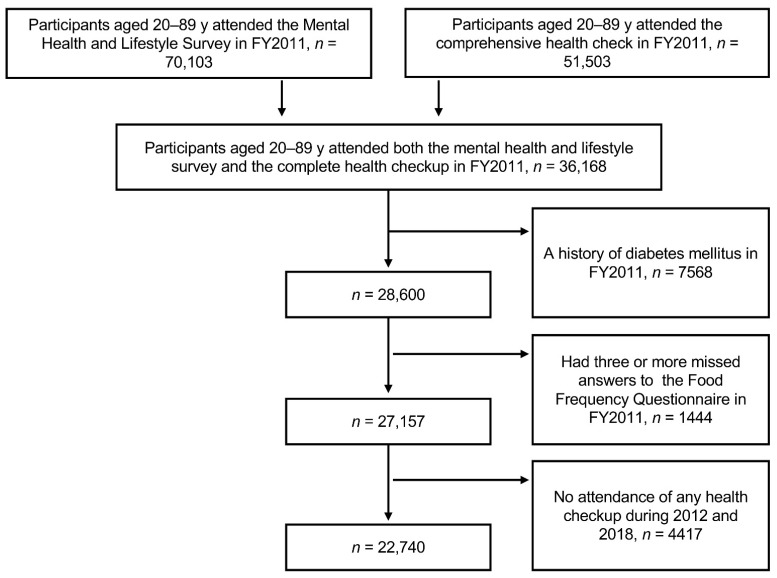
Data flow chart for the analysis of participants of the Fukushima Health Management Survey.

**Figure 2 nutrients-14-04872-f002:**
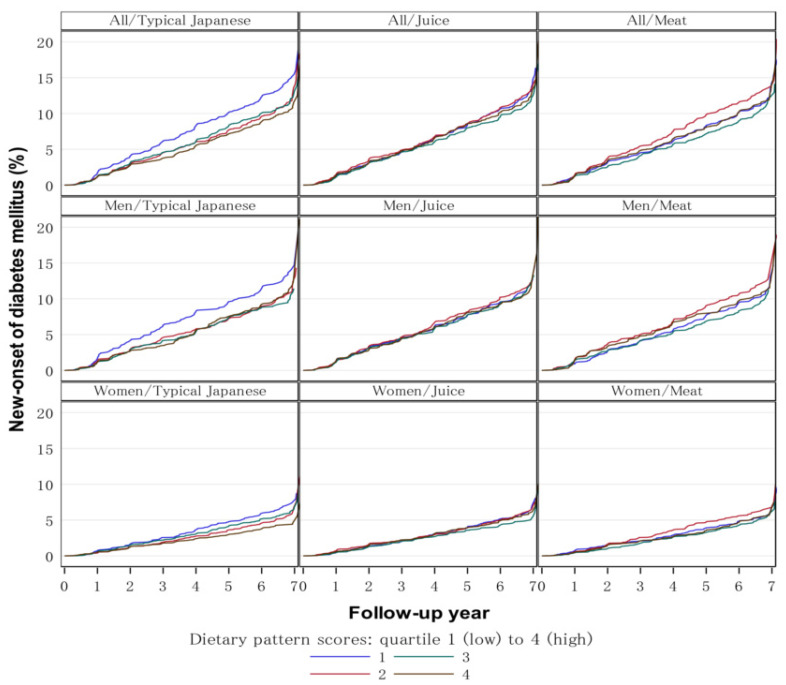
Cumulative incidence of new-onset type 2 diabetes mellitus (T2DM) among dietary patterns in participants of the Fukushima Health Management Survey (FHMS) between 2011 to 2018. The Kaplan–Meier curves were shown in all (upper panel), men (middle panel), and women (lower panel) for typical Japanese, juice, and meat dietary pattern among Q1 (blue lines), Q2 (red lines), Q3 (green lines), and Q4 (brown lines).

**Table 1 nutrients-14-04872-t001:** Characteristics of participants in FY2011, FHMS (*n* = 22,740).

	All (*n* = 22,740)	Men (*n* = 8465)	Women (*n* = 14,275)	*p* Value
Age (years)	55.9 (15.7)	58.2 (15.4)	54.6 (15.8)	<0.001
Education ≥ vocational university	25.8	22.6	27.7	<0.001
Current smoker	16.4	29.4	8.7	<0.001
Current alcohol drinking	45.0	69.8	30.3	<0.001
Physical activity ≥ 2 times/week	34.6	39.0	32.0	<0.001
K6 ≥ 13	13.7	10.6	15.5	<0.001
Live at shelter/temporary house	43.6	43.9	43.5	0.193
BMI (kg/m^2^)	23.4 (3.6)	24.2 (3.3)	22.9 (3.7)	<0.001
BMI ≥ 25 kg/m^2^	29.8	37.6	25.1	<0.001
Hypertension	39.8	49.3	34.1	<0.001
SBP (mmHg)	127.0 (16.9)	130.9 (15.8)	124.7 (17.1)	<0.001
DBP (mmHg)	76.7 (10.9)	79.8 (10.4)	74.8 (10.7)	<0.001
Fasting blood glucose (mg/dL)	93 [88, 100]	96 [90, 103]	92 [87, 98]	<0.001
LDL-C (mg/dL)	124.3 (32.4)	122.8 (32.0)	125.1 (32.6)	<0.001
LDL-C ≥ 140 mg/dL	30.2	29.2	30.8	0.008
HDL-C (mg/dL)	61.3 (15.3)	55.7 (14.4)	64.6 (14.9)	<0.001
HDL-C < 40 mg/dL	5.6	10.3	2.8	<0.001
Triglycerides (mg/dL)	91 [64, 130]	105 [74, 151]	83 [60, 118]	<0.001
Triglycerides ≥ 150 mg/dL	17.8	25.8	13.1	<0.001
Typical Japanese pattern score	−0.02 [−0.71, 0.71]	−0.02 [−0.69, 0.70]	−0.02 [−0.71, 0.71]	0.817
Juice pattern score	−0.18 [−0.69, 0.46]	−0.17 [−0.69, 0.45]	−0.19 [−0.69, 0.46]	0.657
Meat pattern score	−0.21 [−0.67, 0.50]	−0.23 [−0.66, 0.46]	−0.20 [−0.68, 0.53]	0.383

Values are mean (SD), %, or median [IQR]. FHMS, Fukushima Health Management Survey; BMI, body mass index; K6, Kessler Psychological Distress Scale; DBP, diastolic blood pressure; HDL-C, high-density lipoprotein cholesterol; IQR, interquartile range; LDL-C, low-density lipoprotein cholesterol; SBP, systolic blood pressure.

**Table 2 nutrients-14-04872-t002:** Incidences of new-onset type 2 diabetes mellitus (T2DM) during 2012–2018, FHMS.

		2012		2013		2014		2015		2016		2017		2018		Total	Person -Year
Men	**New onset** T2DM	**142**	**(19.4)**	**136**	**(18.6)**	**92**	(12.6)	114	(15.6)	104	(14.2)	83	(11.4)	60	(8.2)	731	40,688
	Fasting blood glucose, ≥126 mg/dL	84	(19.2)	66	(15.1)	51	(11.7)	60	(13.7)	65	(14.9)	52	(11.9)	59	(13.5)	437	41,450
	HbA1c, >6.5%	64	(16.0)	81	(20.3)	42	(10.5)	59	(14.8)	50	(12.5)	56	(14.0)	48	(12)	400	41,558
Women	**New onset** T2DM	113	(15.8)	132	(18.4)	87	(12.1)	114	(15.9)	106	(14.8)	99	(13.8)	66	(9.2)	717	73,082
	Fasting blood glucose, ≥126 mg/dL	59	(15.6)	57	(15.1)	55	(14.6)	52	(13.8)	56	(14.9)	47	(12.5)	51	(13.5)	377	73,946
	HbA1c, >6.5%	52	(12.1)	86	(20.0)	47	(11.0)	61	(14.2)	63	(14.7)	66	(15.4)	54	(12.6)	429	73,854

Values are presented as numbers (%). FHMS, Fukushima Health Management Survey. HbA1c, hemoglobin A1c.

**Table 3 nutrients-14-04872-t003:** Associations between dietary patterns and diabetes mellitus risk, 2011–2018, FHMS.

	Dietary Pattern Scores	All (*n* = 22,740)		Men (*n* = 8465)		Women (*n* = 14,275)	
		HR	95% CI	HR	95% CI	HR	95% CI
**Typical Japanese**							
Model 1 ^a^	Q1 (lowest)	Ref.	-	Ref.	-	Ref.	-
	Q2	**0.79**	**(0.68, 0.92)**	**0.78**	**(0.63, 0.97)**	0.80	(0.64, 1.00)
	Q3	**0.79**	**(0.68, 0.92)**	**0.73**	**(0.58, 0.90)**	0.86	(0.70, 1.07)
	Q4	**0.71**	**(0.60, 0.83)**	**0.78**	**(0.63, 0.97)**	**0.64**	**(0.51, 0.80)**
	*P* for trend	**<0.001**		**0.048**		**<0.001**	
Model 2 ^b^	Q1 (lowest)	Ref.	**-**	Ref.	-	Ref.	-
	Q2	**0.81**	**(0.69, 0.94)**	**0.79**	**(0.64, 0.98)**	0.82	(0.66, 1.03)
	Q3	**0.80**	**(0.69, 0.93)**	**0.72**	**(0.58, 0.90)**	0.89	(0.72, 1.10)
	Q4	**0.74**	**(0.63, 0.86)**	**0.78**	**(0.63, 0.97)**	**0.70**	**(0.56, 0.88)**
	*P* for trend	**0.011**		**0.042**		**0.005**	
Model 3 ^c^	Q1 (lowest)	Ref.	**-**	Ref.	-	Ref.	-
	Q2	**0.82**	**(0.70, 0.96)**	0.81	(0.65, 1.01)	0.84	(0.67, 1.05)
	Q3	**0.83**	**(0.71, 0.97)**	**0.74**	**(0.60, 0.92)**	0.93	(0.75, 1.15)
	Q4	**0.80**	**(0.68, 0.94)**	0.85	(0.68, 1.06)	**0.76**	**(0.60, 0.95)**
	*P* for trend	**0.015**		0.181		**0.04**	
**Juice**							
Model 1 ^a^	Q1 (lowest)	Ref.	-	Ref.	-	Ref.	-
	Q2	1.01	(0.88, 1.17)	1.03	(0.84, 1.27)	1.00	(0.82, 1.23)
	Q3	0.90	(0.78, 1.05)	0.97	(0.79, 1.20)	0.85	(0.68, 1.05)
	Q4	0.96	(0.83, 1.11)	0.97	(0.79, 1.20)	0.96	(0.78, 1.18)
	*P* for trend	0.427		0.690		0.563	
Model 2 ^b^	Q1 (lowest)	Ref.	-	Ref.	-	Ref.	-
	Q2	1.00	(0.86, 1.16)	1.01	(0.82, 1.24)	1.00	(0.82, 1.23)
	Q3	0.89	(0.76, 1.03)	0.95	(0.77, 1.16)	0.84	(0.68, 1.04)
	Q4	0.95	(0.83, 1.11)	0.94	(0.77, 1.16)	0.99	(0.80, 1.21)
	*P* for trend	0385		0.503		0.728	
Model 3 ^c^	Q1 (lowest)	Ref.	**-**	Ref.	-	Ref.	-
	Q2	1.01	(0.87, 1.17)	1.02	(0.83, 1.26)	0.99	(0.81, 1.22)
	Q3	0.90	(0.78, 1.05)	0.97	(0.79, 1.20)	0.83	(0.67, 1.03)
	Q4	0.99	(0.86, 1.15)	0.99	(0.80, 1.23)	1.01	(0.82, 1.24)
	*P* for trend	0.773		0.832		0.912	
**Meat**							
Model 1 ^a^	Q1 (lowest)	Ref.	-	Ref.	-	Ref.	-
	Q2	1.14	(0.99, 1.30)	1.13	(0.94, 1.37)	1.15	(0.95, 1.39)
	Q3	0.89	(0.76, 1.03)	0.89	(0.72, 1.10)	0.89	(0.73, 1.10)
	Q4	1.01	(0.87, 1.17)	1.04	(0.84, 1.29)	0.97	(0.79, 1.20)
	*P* for trend	0.455		0.846		0.415	
Model 2 ^b^	Q1 (lowest)	Ref.	-	Ref.	-	Ref.	-
	Q2	1.13	(0.99, 1.29)	1.12	(0.92, 1.35)	1.15	(0.95, 1.39)
	Q3	0.90	(0.78, 1.05)	0.91	(0.74, 1.13)	0.89	(0.73, 1.10)
	Q4	1.03	(0.88, 1.19)	1.07	(0.87, 1.33)	0.98	(0.80, 1.21)
	*P* for trend	0.694		0.898		0.465	
Model 3 ^c^	Q1 (lowest)	Ref.	**-**	Ref.	-	Ref.	-
	Q2	1.13	(0.99, 1.29)	1.11	(0.91, 1.34)	1.17	(0.96, 1.41)
	Q3	0.91	(0.78, 1.06)	0.90	(0.72, 1.11)	0.92	(0.74, 1.13)
	Q4	1.05	(0.90, 1.22)	1.06	(0.86, 1.32)	1.03	(0.83, 1.27)
	*P* for trend	0.883		0.959		0.747	

Values are HRs (95%CI) or *p*-values. ^a^ Adjusted for age (continuous) and sex; ^b^ Model 1 further adjusted for body mass index (<23, ≥23 to <25, ≥25 kg/m^2^); ^c^ Model 2 further adjusted for smoking (never, former, or current), current drinking (no, yes), education level (<, ≥vocational university), physical activity (<two, ≥two times/week), probable depression (K6 < 13, ≥13), change in residence (no, yes), hypertension (no, yes), and dyslipidemia (no, yes). HR, hazard ratio; CI, confidence interval; FHMS, Fukushima Health Management Survey.

## Data Availability

The datasets analyzed during the present study are not publicly available because the data from the Fukushima Health Management Survey belongs to the government of Fukushima Prefecture and can only be used within the organization.

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
