# Peer review of "Dietary Patterns and New-Onset Type 2 Diabetes Mellitus in Evacuees after the Great East Japan Earthquake: A 7-Year Longitudinal Analysis in the Fukushima Health Management Survey"

_nutrients, 2022, doi:10.3390/nu14224872_

Round 1

Reviewer 1 Report

The manuscript named “Dietary patterns and new-onset type 2 diabetes mellitus in evacuees after the Great East Japan Earthquake: A 7-year longitudinal analysis in the Fukushima Health Management Survey” is meaningful and interesting. While there are some limitations.

Majors:

1. There is lack of explanation of the characteristics of participants either in the background or in the discussion, why do you chose this group to study? What is the difference between this group and general group? What is possible reason which can explain the effect of DP on the occurrence of hyperglycemia? Please add relative content to certain parts.

2. A important variable, that is the change of DP during follow-up, has not put into the regression analysis. Since you have a few times of FFQ, could you supplement it into this study?

3. The description of methodology in dietary record is not clear enough.

Minors:

1. There were some mistakes in Figure 1, such as there was 27156 after excluded those who missed more than three FFQ answers........

2. one-to-four??? percent of the participants who did not answer more than three dietary questions on the FFQ? Please correct it.

3. Please give a blank line between lines 191 and 192. So do as this for other places.

4. There are different description about study period, 2011-2018 or 2012-2018. Please unify them.

5. Please rearrange table 2.

Reviewer 2 Report

Dear authors, I read your article with great interest this study . The manuscript is well structured, and the tables are clear and easy to interpret. The methods are mostly well explained and reproducible, but I sugget to better describe the nutritional assessment. The covariates used in the models are suitable. However, the limitations part of the study needs to be expanded especially regarding the ffq limitations.
